# Machine Learning Approach Using MLP and SVM Algorithms for the Fault Prediction of a Centrifugal Pump in the Oil and Gas Industry

**Pier Francesco Orrù [1], Andrea Zoccheddu [1], Lorenzo Sassu [2], Carmine Mattia [2], Riccardo Cozza [3] and Simone Arena [1,\***

[1] Department of Mechanical, Chemical and Material Engineering, University of Cagliari, Via Marengo 2, 09134 Cagliari, Italy; pforru@unica.it (P.F.O.); and.zoccheddu@gmail.com (A.Z.)

[2] Sartec-Saras Ricerche e Tecnologie Srl, Via 2° Traversa Strada Est, 09032 Cagliari, Italy; lorenzo.sassu@sartec.it (L.S.); carmine.mattia@sartec.it (C.M.)

[3] Saras S.p.A.-S.S. Sulcitana n.195 Km 19, 09018 Cagliari, Italy; riccardo.cozza@saras.it

\* Correspondence: simonearena@unica.it; Tel.: +39-070-6755350

**Abstract:** The demand for cost-effective, reliable and safe machinery operation requires accurate fault detection and classification to achieve an efficient maintenance strategy and increase performance. Furthermore, in strategic sectors such as the oil and gas industry, fault prediction plays a key role to extend component lifetime and reduce unplanned equipment thus preventing costly breakdowns and plant shutdowns. This paper presents the preliminary development of a simple and easy to implement machine learning (ML) model for early fault prediction of a centrifugal pump in the oil and gas industry. The data analysis is based on real-life historical data from process and equipment sensors mounted on the selected machinery. The raw sensor data, mainly from temperature, pressure and vibrations probes, are denoised, pre-processed and successively coded to train the model. To validate the learning capabilities of the ML model, two different algorithms—the Support Vector Machine (SVM) and the Multilayer Perceptron (MLP)—are implemented in KNIME platform. Based on these algorithms, potential faults are successfully recognized and classified ensuring good prediction accuracy. Indeed, results from this preliminary work show that the model allows us to properly detect the trends of system deviations from normal operation behavior and generate fault prediction alerts as a maintenance decision support system for operatives, aiming at avoiding possible incoming failures.

**Keywords:** predictive maintenance; machine learning; artificial neural networks; oil and gas industry; fault diagnosis

## 1. Introduction

In the middle of the fourth industrial revolution, industries are constantly looking for ways to optimize production lines and, at the same time, to reduce their costs. Maintenance expenses typically amount to more than one-third of total operating costs. Traditional maintenance techniques are based on two different strategies: corrective maintenance and preventive maintenance [1]. The former is performed to repair faulty systems and equipment only when a failure has already occurred and, in such a way, the process' direct costs increase. The latter is performed after regular time intervals to prevent systems/equipment failures. Thus, repairs on machinery or components are carried out when they still have an unknown remaining useful life leading to both unplanned downtimes of machinery and an increase in the operating costs. Therefore, a proper maintenance strategy should improve general equipment health status and, furthermore, it should reduce equipment failure rates, minimizing maintenance costs and maximizing equipment's useful life [2]. In this scenario, the concepts

of sustainable energy and environmental management are framed in many industrial activities, since they provide better accountability, better control and costs' allocation, improved performance and waste material reduction [3]. To this end, reliable and efficient maintenance strategies and good practices have to be considered indispensable to improve sustainability, for they increase plant efficiency aiming at both reducing downtime, waste of energy and materials and enhancing socioeconomic well-being significantly [4]. Enhanced by these factors, Predictive Maintenance (PdM) has become one of the hottest topics of the Industry 4.0 era, being based on high quality and optimally scheduled sustainable maintenance practices by integrating physical and digital systems of production environments. Technological developments in recent years have led to a reduction in expenses related to monitoring the health of machinery and to the acquisition and storage of huge amounts of data. The application of cutting-edge analytical models to data provides valuable information and knowledge from manufacturing processes, production systems and equipment to support strategic decision-making.

PdM's goal is to predict a future failure event allowing to react proactively and plan the most appropriate maintenance solution at the right time by using data and algorithms. Thus, through advanced condition monitoring techniques, this approach assures high-value machinery management optimization in highly competitive fields, such as manufacture, transport and energy production. Quite recently, advances in computational performance have allowed the application of machine learning (ML) algorithms which are able to find correlations and to identify patterns over extremely complex and large amounts of data [5,6]. The basic idea is to use machines' data aiming at processing the training database, in order to provide reliable and efficient fault diagnostic models. To date, a large amount of research and studies have been conducted on predictive maintenance using ML algorithms applied to industrial applications. Carvalho et al. [2] presented a Systematic Literature Review (SLR) focused on machine learning methodologies applied to predictive maintenance techniques. Lei et al. [7] presented a SLR on the applications of machine learning to machine fault diagnosis and provided a roadmap for this field. Del Ser et al. [8] presented a detailed review of the recent developments in data analysis strategies and machine learning algorithms for data-driven prognosis within the Industry 4.0 paradigm according to a specific classification in descriptive, predictive and prescriptive models.

According to the results reported in these mentioned SRLs, the number of publications on machine fault diagnosis by using machine learning has rapidly increased since the 2010s. In particular, Del Ser et al. [8] reported more than twenty works on predictive prognostic models but only one of them is based on machine learning modeling focused on oil and gas. In this paper, Costello et al. [9] present a data-driven model for health monitoring of a gas circulator units using real vibration data.

Despite a detailed literature review on machine learning modeling for industrial applications (manufacturing, transportation, energy production, heavy industry among others) it has been difficult to make use of past studies, carried out on real raw data from operating machinery [10–14]. Recently, Hajizadeh [15] and Hanga and Kovalchuk [16] reported the efforts to date of applying artificial intelligence in fault detection in the oil and gas industry, both analyzing its benefits and suggesting ways to ensure its greater adoption for strategic management and technology enablement. Indeed, due to the lack of willingness on the side of companies to make public data ownership, particularly in the field of petrochemical applications, most studies rely on data retrieved from simulations or experimental tests carried out in a laboratory, with ideal setups been used [5,17–19]. In recent years, different ML methods applied to rotating machinery fault diagnosis have been developed. Qian et al. [20,21] proposed a transfer learning method for fault diagnosis of these components under variable working conditions. Su et al. [22], Weiwei et al. [23], Zhang et al. [24], and Lei et al. [25] presented different approaches to develop machine learning algorithms to roller bearing fault diagnosis. Bilski [26] presented a Support Vector Machines (SVM) model to monitor the health state of the induction machine. Jirdehi and Rezaei [27] proposed two methods based on artificial neural networks (ANN) and adaptive neuro-fuzzy inference system (ANFIS) to estimate the induction motor parameters in the single-cage and double-cage models. Romeo et al. [28] proposed an innovative Design Support System (DesSS)

for the prediction and estimation of machine specification data. Giantomassi et al. [29] investigated the fault detection and diagnosis of an electric motor by using an estimation algorithm of the probability density function of current signals. Paolanti et al. [30] presented a ML architecture based on Random Forest approach for fault diagnosis of a cutting machine. Yang et al. [31] presented a fault diagnosis scheme combined with hierarchical symbolic analysis (HSA) and convolutional neural network (CNN). They focused the analysis on experimental testing of a centrifugal pump and a motor bearing. Pang et al. [32] proposed a new fault diagnosis method based on multiple-domain data by using experimental testing dataset of gearbox, rotor and engine rolling bearing. Chen et al. [33] proposed a novel fault diagnosis approach integrating CNN and Extreme Learning Machine (ELM) by using experimental dataset of gearbox and motor bearing. Zhang et al. [34] presented an ant colony algorithm for synchronous feature selection and parameter optimization for support vector machine in fault diagnosis of rotating machinery by the experiment of rotor system and locomotive roller bearings. Wang et al. [35] developed a novel ensemble ELM network for compound-fault diagnosis of rotating machinery by using real-world database. Panda et al. [36] focused on the vibration-based condition monitoring and fault diagnosis of centrifugal pumps using the Support Vector Machine (SVM) algorithm. Wang et al. [37] proposed a novel fault diagnosis method that utilizes ensemble learning with differentiated probabilistic neural networks by using an experimental dataset of rotary actuator systems and plunger hydraulics. Liu et al. [38] presented a comprehensive review of artificial intelligence algorithms in rotating machinery fault diagnosis, from both the views of theory background and industrial applications. Tang et al. [39] proposed a novel adaptive learning rate deep belief network combined with Nesterov momentum for rotating machinery fault diagnosis. The authors implemented the proposed model on datasets from gearbox and locomotive bearing test rigs. Yu et al. [40] proposed a probabilistic neural network algorithm to predict oil-immersed transformer internal faults. Zenisek et al. [41] reported a machine learning-based approach for detecting drifting behavior to identify wear and tear, and consequent malfunctioning by analyzing real time condition monitoring data. Li et al. [12] proposed a machine learning approach to predict impending failures and alarms of critical rail car components aiming at both driving proactive inspections and repairs and reducing operational equipment failure. Lee et al. [42] presented an algorithm based on artificial intelligent for predictive maintenance to monitor two critical machine tool system elements, i.e., the cutting tool and the spindle motor. Guedes et al. [43] investigated the evaluation of the electrical insulation of the stator of three-phase induction motors and the classification of the failure mechanism using artificial neural networks (ANNs). The literature review presented above highlights that machine learning has attracted both academic and practitioners in the field of industrial maintenance management, emerging as a powerful tool for intelligent predictive algorithms development in many applications. ML approaches are able to find patterns of interest based upon the data monitored from the process or asset and transform raw data into features spaces by means of mathematical algorithms. To this end, the recent development in technological fields—such as Cyber-Physical systems, Internet of Things (IoT) and Big Data—has provided a large amount of data extracted from production processes, allowing ML techniques to face a wide variety of problems including prediction, anomaly detection, classification, regression, or forecasting [44]. On the other hand, as reported in [45], the implementation of ML techniques in real factories remains rather challenging. Major challenges are:

- Data acquisition and storage [46]. A large amount of data is used in the process of training and learning. In general, more data leads to more reliable models and, consequently, better results. However, data should be representative of the analyzed process. Therefore, the acquisition of relevant data has a strong influence on the ML algorithm performance. Moreover, the processing of large repositories of time series data capturing during the process is necessary to handle them aiming at extracting valuable knowledge and information;

- Selection and design of the ML algorithm [44]. The ML model must be able to estimate machinery condition in a short time interval, aiming at performing proper and agile decision-making. As said before, the growing interest in the field of ML in manufacturing has led to the development

of a large number of different ML algorithms or at least variations to adapt their operation according to different situations requiring deep skills and expertise. Although the core principles of ML are now accessible to a wider audience, adequate knowledge and deep technical know-how are required.

The foregoing discussion suggests that there is a need to address these challenges in order to enable an effective implementation of ML techniques aiming at leading to a good maintenance strategy for an accurate fault diagnosis. In this work, a fault prediction technique based on a ML supervised method using data from actual working operating conditions is presented. The aim is to develop a ML model that is ready to implement. The motivation behind this approach is based on the need for companies to have an accurate, easy and responsive tool in order to predict the failures of one or more system equipments. This need is not easy to meet, especially when it comes to complex problems that require the handling of large number of computations.

To select a simple and accurate ML model to failure prediction, a comparison between two different algorithms, the Support Vector Machine (SVM) and the Multilayer Perceptron (MLP) is performed. The open-source software KNIME platform is adopted to set-up the model workflow. The data used in this study come from a real oil and gas industry operating centrifugal pump inserted inside the production line of the SARLUX refinery (Sarroch, Italy). The data analysis is complicated by difficulties linked to the production processes, such as the non-rigorous recording of events by field operators and sensors' malfunction. The following chapters are structured as follows: Section 2 illustrates the methodologies applied to the case study. Particularly, the different steps performed to develop SVM and ANN models and selection of the most suitable input variables are briefed; Section 3 reports the results achieved for both the selected algorithms and, finally, in Section 4, conclusions and future research are depicted.

## 2. Methodology Applied to the Case Study

Machine learning is a subfield of computer science which can be defined as the process of solving a practical problem by gathering a dataset and algorithmically building a statistical model based on that dataset to extract information about it [47]. Learning can be supervised, semi-supervised, unsupervised and by reinforcement. In this paper, a supervised learning technique has been used, since every sample of the dataset has been labeled to train a model that takes feature vectors as input and outputs information that allows to deduce the label of new collected data.

The Predictive Maintenance (PdM) oriented pipeline built for this study relies on machine learning techniques applied to data analysis collected upon operations of a centrifugal pump. The pipeline has been structured as shown in Figure 1. The reported pipeline represents the schematic diagram of the ML technique which involves some steps: (i) data acquisition step, in which data are collected and stored aiming at achieving relevant data to equipment health; (ii) data processing step, in which collected data have been labeled and cleaned in order to be efficiently processed by the ML model. Furthermore, feature engineering has been performed to extract new features and the relevant ones have been selected to train the model; (iii) model selection, training and validation step, in which the selection of the proper ML model is selected, trained and then validated by maximizing the more appropriate evaluation metrics.

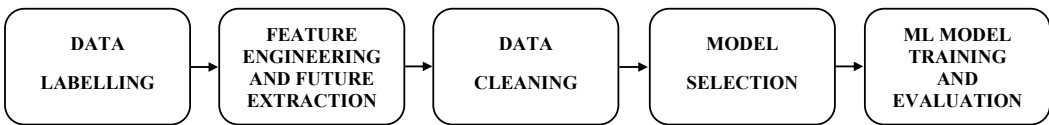

**Figure 1.** Generic schematic diagram of the proposed method for machinery fault diagnosis.

Knime platform is used to design the workflow of centrifugal pump fault diagnosis, as reported in Figure 2. In accordance with the logic of this workflow, the data analysis and processing will be

briefly described in the following. A supervised predictive maintenance-oriented problem can mainly be addressed by means of two approaches:

- A classification approach aims at predicting whether the centrifugal pump will fail or not within a future time interval, called failure prediction horizon T in this paper, whose length is determined by various factors, such as management needs and possibilities, domain expert decisions, etc. The parameter T is defined as the minimum lead time required to replace components before a problem occurs and its value is defined in such a way that it would be possible to apply proactively maintenance to avoid the problem.
- A regression approach aims at predicting how much time is left before the next failure event (RUL, Remaining Useful Life prediction) [48].

This study applies a classification technique. The parameter T has been set to be a one-week time interval, since domain experts have identified it as the minimum time required to apply maintenance to the pump before a failure event. In order to perform classification, we need to categorize each timestamp in a class using arbitrary labels. Two classes have been identified to define a categorical variable named "Current pump operating status": a positive instance class "1" referring to the samples included inside the failure prediction horizon and a negative instance class "0" referring to normal operating conditions. Therefore, in order to correctly classify the failure prediction horizon and act before it happens, 168 timestamps before each failure event have been labeled as "1", the remaining as "0". It is important to notice that the definition of the parameter T implies the conservative hypothesis that in that period, a change from normal operating conditions of the pump happens. According to this consideration, data immediately prior to T could be seen as noise, since it would be difficult to be considered belonging to one of the classes, and will be removed from the analysis further on.

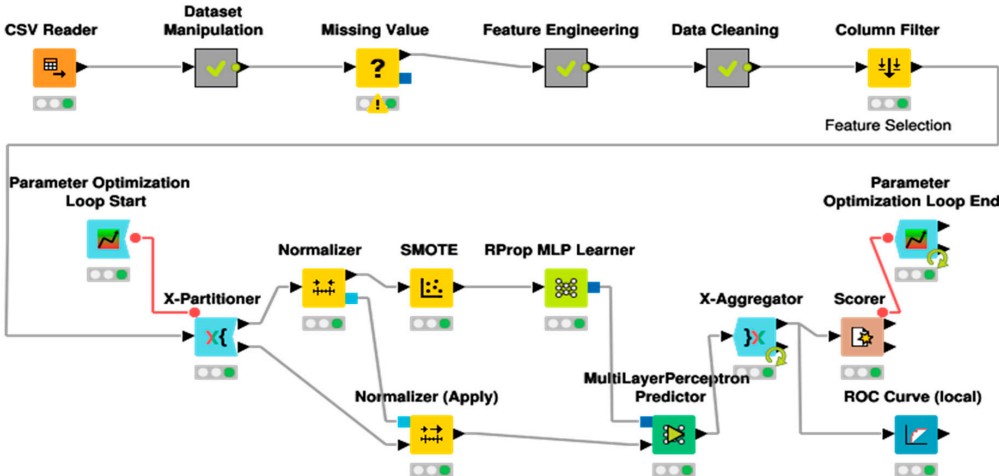

**Figure 2.** Knime workflow built for fault diagnosis of a centrifugal pump.

### 2.1. Data Acquisition Step

Dataset

The dataset used in this work refers to centrifugal pump operations-related sensor readings which were collected between January 2010 and May 2014 since, on that date, the pump was replaced.

The data for PdM are in time series format. Dataset includes a timestamp, a set of sensor readings collected at the same time as timestamps, and device identifiers.

Measurements from eight different sensors have been included in the analysis and used to build the features, i.e., the predictive attributes the model is built on: one of flow rate, two of bearings vibration, two of axial displacement and three of motor coil temperature.

Sampling has been executed with a frequency of one sample per hour, in order to deal with a reasonable amount of data.

By referring to machine's historical events register, it has been possible to identify eight major failure events which occurred during the considered period, four of which were related to seal leakages and four to ambiguously registered failure events. The lack of standardized recording and reporting failure methodologies leads to the need to reject such events due to uncertain data interpretation. Thus, in this study, only four seal leakages failure events have been included in the dataset.

### 2.2. Data Processing Step

#### 2.2.1. Missing Values Treatment

For each feature, it has been necessary to fill gaps between data samples due to missing values. Missing values are a common issue related to on-field operating machinery measurements, since many ML algorithms are not efficient in dealing with datasets affected by missing data. This has been performed by means of the "Missing Values" node in KNIME (Figure 2), using the linear interpolation technique, which replaces each missing value with new ones obtained by linear interpolation between the previous and next non-missing value encountered for each feature column. Linear interpolation was chosen, among different imputation methods, to be simple and effective for time series data [49].

#### 2.2.2. Feature Engineering

The correlation between a data sample and preceding samples in chronological order is one of the most important aspects of a time series type dataset. It is appropriate to perform a feature engineering in order to introduce this prior knowledge in the ML algorithm. Thus, eight new features have been created by means of backward simple moving average calculation, based upon a ten hours window over each data point for each feature. This step, achieved with the "Moving Average" node in KNIME, guarantees to take the correlation existing between a sample and the preceding ones of the time series into consideration and ensures noise attenuation.

#### 2.2.3. Data Cleaning

It has also been necessary to filter out the data related to downtime periods, in which the machine has been shut down due to maintenance operations or other issues, as well as data related to start-up periods. This has been done by setting up a threshold over the flow rate sensor, based upon domain knowledge of the process the pump is inserted in: every sample corresponding to a flow rate below the threshold has been filtered out by means of a "Rule Based Filter" node. Model's performance would have been highly penalized by including this set of data in the study, since it does not refer to neither a normal operating condition nor a condition included in the failure prediction horizon interval.

It has been considered appropriate to eliminate all the data following a pump substitution, which may have caused variations of normal operating conditions behavior from the ones of previously installed machine.

A further data filtration has been performed in order to wipe out samples included in the time interval between one week and three weeks prior to each failure event, due to the high level of uncertainty related to labeling this sets of data either as normal operating conditions or pre-failure ones and in order to get a more class-balanced dataset.

### 2.3. Model Selection and Validation

The machine learning model hyperparameters have been optimized by means of a grid search and k-fold cross validation technique. The results have been validated by means of the k-fold cross validation technique to assess how accurately the model would perform in practice and to

avoid overfitting. The input dataset is partitioned over a loop in k equal-sized subsets: of all subsets, a single one is retained as validation set and the remaining as training for the machine learning algorithm. The loop executes k iterations, each one referring to a different validation set and, accordingly, a different training set. The scoring metrics of the model are given by the average of the scoring metrics of all iterations [50].

The value of k parameter has been chosen in such a way that every fold of the k equal-sized subsets includes at least one failure event and data samples preceding a failure event are not split between folds. Since the dataset's failure events are not distributed uniformly, an optimal value for k has been determined. In the case study, a value of three for k ensures a good representation of the input dataset by the training and validation subset groups.

Since the algorithm's performance would have been penalized by the highly unbalanced dataset, as it will be discussed in the Section 2.3.2, the SMOTE (Synthetic Minority Oversampling Technique) algorithm [51] has been used. This approach is inspired by a technique developed in [52] characterized by the implementation of extra training data by performing certain operations on real data. A dataset is imbalanced if the classification categories are not approximately equally represented. Thus, the SMOTE algorithm increases the number of samples of the minority class by creating new synthetic observations generated through simple mathematical and geometric operations. In this case, the number of samples of the minority class is set in order to equalize its size with the majority class.

### 2.3.1. Classification Algorithms

In this work, two widely used classification techniques are implemented and compared, namely: Support Vector Machines (SVMs) and Multilayer Perceptron (MLP).

Support Vector Machines (SVMs): it is a non-probabilistic binary (two class) linear classifier originally proposed by Vapnik and Cortes [53,54]. SVM separates data across a decision boundary, named hyperplane $f(x) = 0$, by solving a constrained quadratic optimization problem based on the structural risk minimization. The given data input $x_i$ $(i = 1, 2, \ldots, N)$ consists of objects with different labels corresponding to the two classes namely positive and negative class. In the case of linearly data, the hyperplane that separates the given data is obtained by Equation (1):

$$y = f(x) = W^T x + b = \sum_{i=1}^{N} W_i x_i + b \tag{1}$$

where N is the number of the samples, W is a N-dimensional vector and b is a scalar. The vector W and scalar b are used to define the position of separating hyperplane. The optimal separating hyperplane is the separating hyperplane that creates the maximum distance between the plane and the nearest data, i.e., the maximum margin. SVM can also be used in non-linear classification tasks with the application of the kernel function. Indeed, working in the high-dimensional feature space generates problems due to handling non-linearly separable features which can be solved by using the kernel function. The selection of the proper kernel function is very important, since it defines the feature space in which the training dataset will be classified. In this paper, the kernel Radial Basis Function (RBF) is used. The RBF kernel hyper parameter $\gamma$ and the SVM penalty parameter C are optimally selected to obtain the best classification performance. In this work, the values of these two parameters are $\gamma = 0$ and $C = 55$.

Multilayer Perceptron (MLP) has been used as an Artificial Neural Networks (ANN) classification algorithm. Since MLP requires normalized data as input, z-score normalization has been made prior to the algorithm training by means of a "Normalizer" node. Then, the same technique has been applied to test data. MLPs are a powerful class of nonlinear statistical models which consist of multiple layers of nodes in a directed graph, with each layer fully connected to the next one. There are three different type of layers, i.e., input, hidden and output layer. Thus, except for the input nodes, each node is a neuron (or processing element) with a nonlinear activation function [50].

Given data input $x_i$ $(i = 1, 2, \ldots, N)$, the neural model output $y$ can be obtained by Equation (2):

$$y = f\left(W^T x\right) = f\left(\sum_{i=1}^{N} W_i x_i + b\right) \tag{2}$$

where f is the activation function, N is the number of the neurons, W are the ANN model weights and *b* is the bias vector.

A binary classification MLP's output is a value included in the interval between 0 and 1, which could be considered as the probability of the positive target class. A parameter optimization loop has been used to tune the hyperparameters of the model so that precision and recall over class "1" are maximized, as it will be discussed in the Section 2.3.2. The model selection has been performed by optimizing the number of hidden layers and the number of hidden neurons per layer of MLP. In this work, three hidden layers composed of ten neurons for each layer are adopted.

MLP was trained using the resilient propagation algorithm (RPROP). RPROP is a first-order neural network optimization algorithm, developed to train shallow MLP in a fast and robust way. Unlike other optimization algorithms, it has no hyperparameter to choose from [55], this leads to a significant reduction in the required computational resources. The algorithm implementation has been performed by means of the "RProp MLP" node.

### 2.3.2. Model Evaluation Metrics

As usual, a predictive maintenance-oriented binary classification problem based on real operating machinery data suffers from the highly unbalanced distribution of the classes over the samples. Failure is a very rare event in the dataset, so only less than 1% of original data belong to class "1" and more than 99% to class "0".

Machine learning algorithms do not work properly with this type of dataset, neither do standard evaluation metrics. In this regard, it is important to emphasize the fact that overall accuracy is a useless metric of model evaluation for this type of study, although it is the most commonly used evaluation metrics for classification models.

True positives (TP) and true negatives (TN) are outcomes of the positive class and negative class, respectively, correctly classified by the model, whereas false positives (FP) and false negatives (FN) are incorrectly classified outcomes. Overall accuracy (OA) is defined by the Equation (3):

$$\text{Overall Accuracy} = \frac{(\text{TP} + \text{TN})}{(\text{TP} + \text{FP} + \text{TN} + \text{FN})} \tag{3}$$

Mediated between the two classes, so it is possible for the model to achieve a nearly perfect overall accuracy by always predicting the majority class.

For highly unbalanced datasets, the best way to judge a model is to use per-class evaluation metrics such as precision, recall, F1 score (weighted average of precision and recall), expressed as in Equations (4)–(6), respectively:

$$\text{Precision} = \frac{\text{TP}}{(\text{TP} + \text{FP})} \tag{4}$$

$$\text{Recall} = \frac{\text{TP}}{(\text{TP} + \text{FN})} \tag{5}$$

$$\text{F1 score} = \frac{2 \cdot (\text{Presision} \cdot \text{Recall})}{(\text{Precision} + \text{Recall})} \tag{6}$$

Additional significant parameters adopted are Cohen's Kappa and Area under the curve (AUC) of the ROC (receiver operating characteristic) Curve.

Since the goal of the study is to predict whether the machine will fail or not in a future interval, recall and precision over the class "1" are the evaluation metric chosen to be maximized.

Furthermore, machine learning models training for predictive maintenance-oriented failure prediction requires a clear understanding of the business requirements and the tolerance to false positives and false negatives. Some businesses could afford having a high number of the former whilst some others of the latter, thus the optimization loop aims at maximizing the metric associated with the requirements.

## 3. Results

The "Scorer" node outputs confusion matrix and evaluation metrics statistics tables. The performance of the two adopted algorithms is evaluated by a confusion matrix as illustrated in Tables 1 and 2 for SVM and MLP algorithm, respectively, where the columns are the predicted class and the rows are the actual class.

**Table 1.** Confusion matrix of the adopted dataset using Support Vector Machine algorithm.

|         | Class 0 | Class 1 |
|---------|---------|---------|
| Class 0 | 27,745  | 71      |
| Class 1 | 456     | 175     |

**Table 2.** Confusion matrix of the adopted dataset using Multilayer Perceptron algorithm.

|         | Class 0 | Class 1 |
|---------|---------|---------|
| Class 0 | 27,684  | 132     |
| Class 1 | 366     | 265     |

### 3.1. Results—Support Vector Machine Algorithm

Table 3 reports the evaluation metric for SVM algorithm. It can be noticed that a high value of overall accuracy equal to 98.1% has been achieved. High values of precision over class "0" and class "1" have been obtained which corresponds to the portion of positive and negative correct classifications, respectively. On the other hand, the value of recall over class "1", which is the percentage of actual positives identified correctly, is quite low. This means that the algorithm has been penalized by the high number of false negatives such that it affects the fault predictive performance of the ML model.

**Table 3.** Evaluation metric for Support Vector Machine algorithm.

|         | Recall | Precision | F1 Score | Accuracy | Cohen's K |
|---------|--------|-----------|----------|----------|-----------|
| Class 0 | 99.7%  | 98.4%     | 99.1%    | -        | -         |
| Class 1 | 27.7%  | 71.1%     | 39.9%    | -        | -         |
| Overall | -      | -         | -        | 98.1%    | 0.392     |

Figure 3 reports the timeline of both the actual and predicted classes during the selected period of collected data. The first row of the timeline represents the predicted values obtained by using the SVM algorithm, while the second row of the timeline represents the data collected from the sensors mounted on the centrifugal pump. The figure shows that the selected algorithm allows us to predict only one of the four failures that occurred and did not produce a false positive.

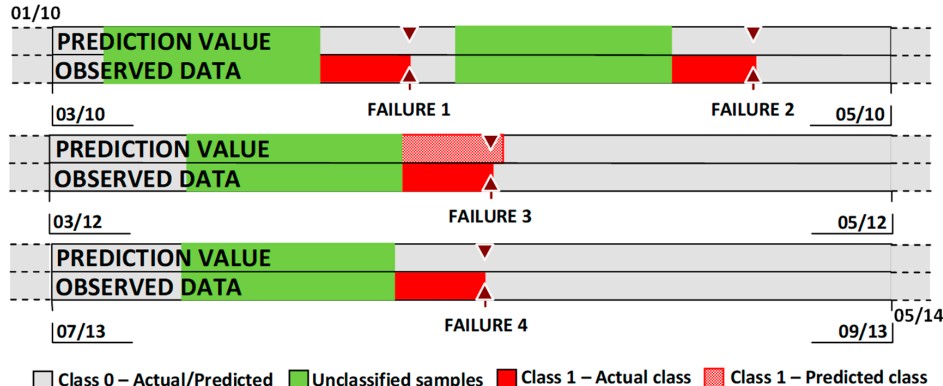

**Figure 3.** Timeline of actual and predicted classes during the selected analysis period using SVM algorithm.

## 3.2. Results—Multilayer Perceptron Algorithm

By analyzing Table 4, it can be noticed that an overall accuracy of 98.2% has been achieved. Although this value is slightly higher than that obtained with SVM algorithm, a lower value of precision over class "1" is achieved. On the other hand, the use of MLP algorithm provides a high value of recall over class "1" equal to 42.2%, which corresponds to the percentage of actual positives identified correctly. These metrics results are computed by using a standard threshold upon the MLP output probability equal to 0.5. Finally, by comparing the two cases, it can be noticed that a higher value of the Cohen's Kappa is achieved by using MLP algorithm. In this case, Cohen's Kappa is 0.507 and it represents the extent to which the prediction values are correct representations of the real observed data used.

**Table 4.** Evaluation metric for Multilayer Perceptron algorithm.

|          | Recall | Precision | F1 Score | Accuracy | Cohen's K |
|----------|--------|-----------|----------|----------|-----------|
| Class 0  | 99.5%  | 98.7%     | 99.1%    | -        | -         |
| Class 1  | 42.2%  | 66.8%     | 51.6%    | -        | -         |
| Overall  | -      | -         | -        | 98.2%    | 0.507     |

In this case too, the timeline of both the actual and predicted classes during the selected period of collected data is reported in Figure 4. The MLP algorithm allows predicting two of the four failures that occurred in the selected period providing better predicting performance than SVM algorithm. On the other hand, false positive predicted values are detected. This means that the algorithm has also identified a fault during the normal operating conditions of the machinery.

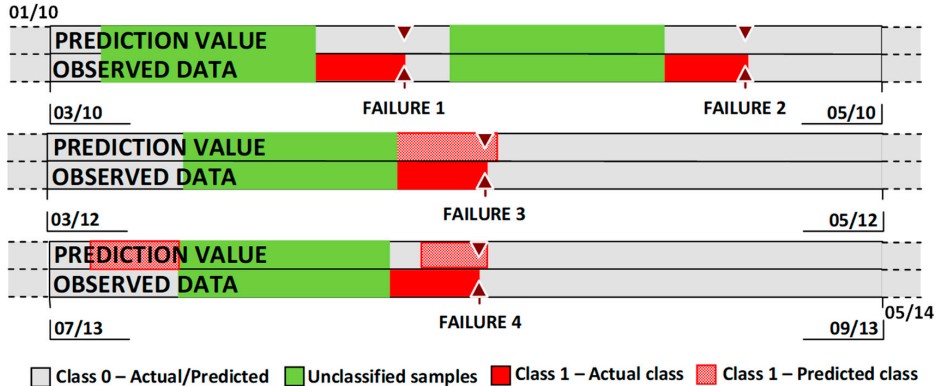

**Figure 4.** Timeline of actual and predicted classes during the selected analysis period using MLP algorithm.

An ROC Curve is another commonly used method of visualizing the performance of a binary classifier. In this work, the ROC curve of the MLP, our best classifier, is reported. In this case, it results from varying the decision threshold and plotting the true positive rate against the false positive rate (see Figure 5).

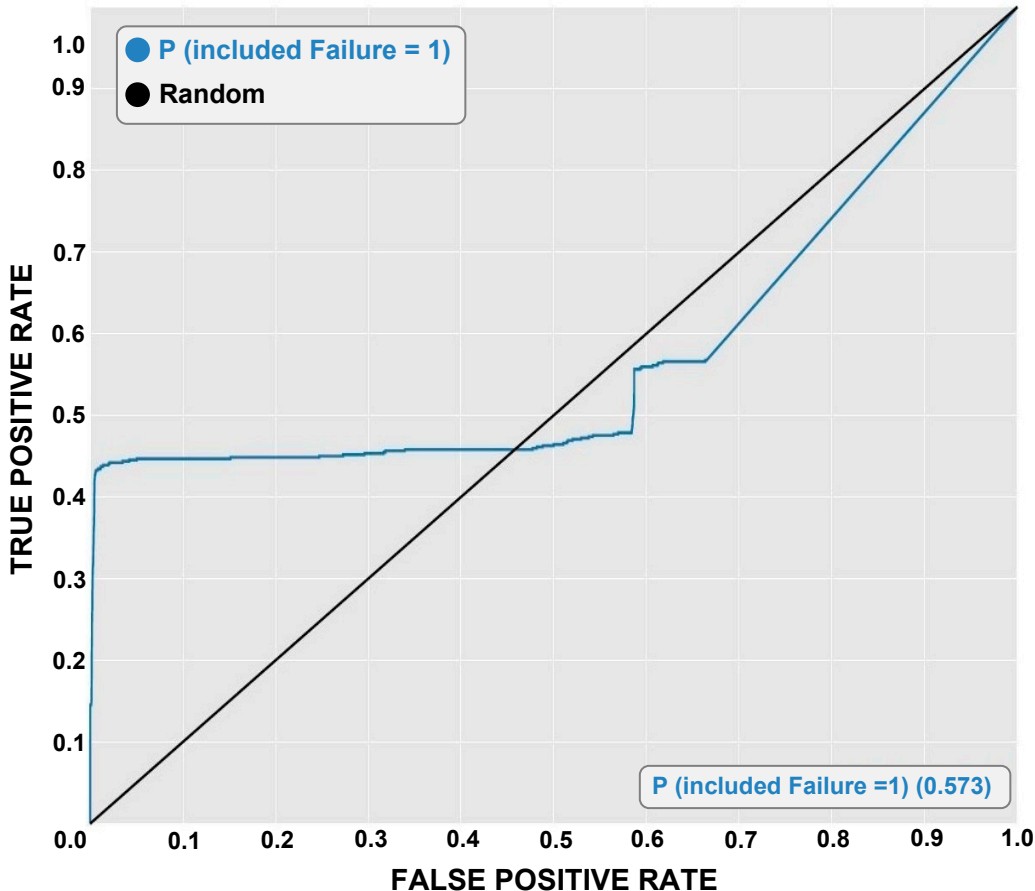

**Figure 5.** ROC Curve for the Multilayer Perceptron algorithm.

Analyzing the ROC Curve, as illustrated in Figure 3, the AUC (Area Under Curve) results equal 0.5726; while it is possible to notice that by changing the threshold value, the FP number could be reduced without reducing the TP number.

All these obtained metrics values are considered satisfactory when taking various factors into consideration:

- The failure events register results incomplete and un-standardized.
- The dataset includes just four non-ambiguous fault events, which is a very small number. Consequently, the dataset is highly unbalanced between classes and the algorithm training set contains very few examples of class "1". By using the SMOTE algorithm, classes are balanced—but still the information quality provided by a "synthetic" sample is not of as high a quality as is provided by real data.
- Inadequate sensory equipment, since the machine parts are not equipped with the most appropriate sensors.
- Sampling frequency has been set to one sample per hour, providing a smaller amount of data to be stored and analyzed.

While limitations inducted by the first three factors are difficult to overcome in a short term, a higher sampling frequency is easily achievable and would result in a larger amount of data, which could

improve ML algorithm performances. Moreover, data cleaning and feature extraction and selection processes will be improved.

## 4. Conclusions

This paper proposes the preliminary development of a machine learning-supervised algorithm for the fault diagnosis of rotating machinery in the oil and gas industry. The basic idea is to develop a simple and easy to implement ML model aiming at performing agile and informed decision-making.

The data come from a real operating centrifugal pump that works within the production line of SARLUX refinery (Sarroch, Italy). Eight different sensors have been used to build the features: one of flow rate, two of bearings vibration, two of axial displacement, and three of motor coil temperature. The adoption of a real dataset strongly influences further decisions of techniques to use for feature engineering, data labeling and machine learning techniques. Two different algorithms are used and compared—the Support Vector Machine (SVM), and the Multilayer Perceptron (MLP).

The main purpose of this work is not based on finding a highly and extremely accurate ML model, but rather to show how, with a simple and intuitive ML algorithm, it is possible to have good forecast results. Indeed, the results show that the proposed algorithms present good overall classification performances, achieving a good capability to identify the health status of the monitored machine. The SVM algorithm shows higher precision than the MLP but lower recall over positive class, while MLP shows better classification performance, predicting two of the four failures that occurred in the selected period. Thus, based on these promising algorithm assessments, a larger scale experiment series is planned, aiming at reducing maintenance costs by the optimal scheduling of sustainable maintenance actions and increasing the life expectancy of the centrifugal pump. To this end, availability-based maintenance planning procedures and the associated costs might be another worthwhile step. Future work will be aimed at improving the overall performance metric by both having more robust data-set and considering a different set of features to improve and make the results generally applicable—or at least reduce possible bias. Finally, additional work is needed to validate the proposed method by considering different industrial components.

**Author Contributions:** Conceptualization, P.F.O. and R.C.; methodology, S.A., A.Z. and C.M.; validation, A.Z., C.M. and L.S.; investigation, S.A. and A.Z.; data curation, L.S.; writing—original draft preparation, S.A., A.Z., and C.M.; writing—review and editing, All Authors; supervision, P.F.O., L.S. and R.C. All authors have read and agreed to the published version of the manuscript.

**Funding:** This research received no external funding.

**Acknowledgments:** Zoccheddu A. gratefully acknowledges Saras Ricerche e Tecnologie Srl for the internship opportunity. The authors wish to thank SARLUX Refinery (Sarroch, Italy) for kindly supplying the data used in the paper.

**Conflicts of Interest:** The authors declare no conflict of interest.

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
