# Peer review of "Machine Learning Approach Using MLP and SVM Algorithms for the Fault Prediction of a Centrifugal Pump in the Oil and Gas Industry"

_sustainability, doi:10.3390/su12114776_

Round 1

Reviewer 1 Report

I would like to congratulate the Authors on the significant progress in the manuscript. I accept the paper and in the same time believe that before final submission the Authors would consider two suggestions given below. Great work, by the way!

A language suggestion - English speaking world does not like the word "but" - it is suggested to change it into "however" or any other word formation.

Future research should be add in conclusion part, e.g. it would be worth applying simulation methods for study on availability of centrifugal pump as e.g. in https://doi.org/10.26552/com.C.2020.2.107-114

Author Response

First of all, we thank Associated Editor and Reviewer for their time and effort and very valuable comments.

Reviewer 1:

I would like to congratulate the Authors on the significant progress in the manuscript. I accept the paper and in the same time believe that before final submission the Authors would consider two suggestions given below. Great work, by the way!

  • A language suggestion - English speaking world does not like the word "but" - it is suggested to change it into "however" or any other word formation.

Reply: The manuscript was proofread to eliminate grammar errors, typos and to improve English in the manuscript.

  • Future research should be add in conclusion part, e.g. it would be worth applying simulation methods for study on availability of centrifugal pump as e.g. in https://doi.org/10.26552/com.C.2020.2.107-114

Reply: To take into account the comment, the content of the conclusion part was reworked as suggested by the reviewer. Further clarifications were added in lines 474-479.

“To this end, availability-based maintenance planning procedures and the associated costs might be another worthwhile step. Future work will be aimed at additional work is needed both to improve improving the overall performance metric by both having more robust data-set and considering a different set of features to improve and make generally applicable the results or at least reduce possible bias. Finally, additional work is needed and to validate the proposed method by considering different industrial components.”

Reviewer 2 Report

The authors addressed my major comments. I suggest that the authors should carefully revise the grammar and check the resolution of the figure (i.e. figure 2 and figure 5).

Author Response

First of all, we thank Associated Editor and Reviewer for their time and effort and very valuable comments.

Reviewer 2:

The authors addressed my major comments.

  • I suggest that the authors should carefully revise the grammar and check the resolution of the figure (i.e. figure 2 and figure 5).

Reply: The manuscript was proofread to eliminate grammar errors, typos and to improve English in the manuscript. Moreover, the required modification of figures 2 and 5 are done aiming at improving their quality and resolution.

This manuscript is a resubmission of an earlier submission. The following is a list of the peer review reports and author responses from that submission.

Round 1

Reviewer 1 Report

The paper presents a preliminary development of a machine learning (ML) model for early fault prediction of a centrifugal pump in Oil&Gas industry. The raw sensor data were collected from temperature, pressure and vibrations signals. The Support Vector Machine (SVM) and MUltilayer Perceptron (MLP) were employed as the standard ML model for predicting potential faults.

Although the application-domain is interesting I have several concerns mainly due to the lack of a novel contribution with respect to the state-of-the-art, the lack of a clear organization and explanation about the proposed methodology and the lack of satisfactory results (in terms of TPR). I summarize my major concerns below:

Section 1: The authors claimed that "the literature on ML modeling using real raw data is relatively limited". I suggest that the authors should improve the literature review by including and citing the following papers: [1] Bilski, P. (2014). Application of support vector machines to the induction motor parameters identification. Measurement, 51, 377–386. [2] Krings, A., Cossale, M., Tenconi, A., Soulard, J., Cavagnino, A., & Boglietti, A. (2017). Magnetic materials used in electrical machines: A comparison and selection guide for early machine design. IEEE Industry Applications Magazine, 23(6), 21–28. [3] Jirdehi, M. A., & Rezaei, A. (2016). Parameters estimation of squirrel-cage induction motors using ANN and ANFIS. Alexandria Engineering Journal, 55(1), 357– 368. [4] L Romeo, J Loncarski, M Paolanti, G Bocchini, A Mancini, E Frontoni, Machine learning-based design support system for the prediction of heterogeneous machine parameters in industry 4.0, Expert Systems with Applications 140, 112869 [5] A. Giantomassi, F. Ferracuti, S. Iarlori, G. Ippoliti and S. Longhi, "Electric Motor Fault Detection and Diagnosis by Kernel Density Estimation and Kullback–Leibler Divergence Based on Stator Current Measurements," in IEEE Transactions on Industrial Electronics, vol. 62, no. 3, pp. 1770-1780, March 2015. [6] Paolanti, M., Romeo, L., Felicetti, A., Mancini, A., Frontoni, E., & Loncarski, J. (2018, July). Machine learning approach for predictive maintenance in industry 4.0. In 2018 14th IEEE/ASME International Conference on Mechatronic and Embedded Systems and Applications (MESA) (pp. 1-6). IEEE. Please clarify the main difference with respect to the above mentioned literature Section 2: it is not clear how the authors model the temporal evolution of the historical data. What is the rationale that "timestamps before each failure event have been labeled as “1”, the remaining as “0”?  Do the authors apply other data imputation techniques? Please provide additional proof about the benefit of linear interpolation. The authors should better explain the feature engineering step. I expect to find at least features extracted in the time and frequency domain (e.g. for vibration signals).  I suggest that the authors should provide additional information about the application of SMOTE. What is the percentage of performed oversampling? How do the authors optimize the hyperparameters of MLP and SVM? There is too much background about SVM and MLP.  It seems that the novel contribution is very poor. I suggest that the authors should emphasize what are the main advantages of the proposed pipeline (related to the fault diagnosis prediction) with respect to other competitors. Section 3: the results of the proposed algorithm are not at all satisfactory. The authors should hardly revise the proposed methodology (mainly focus on feature extraction and the machine learning model). 

Reviewer 2 Report

The stability of the information provided in the article is very debatable.

It would be worth to include short introduction to Industry 4.0, e.g. based on Kostrzewski M., Varjan P., Gnap J. (2020) Solutions Dedicated to Internal Logistics 4.0. In: Grzybowska K., Awasthi A., Sawhney R. (eds) Sustainable Logistics and Production in Industry 4.0. EcoProduction (Environmental Issues in Logistics and Manufacturing). Springer, Cham, pp 243-262. ISBN 978-3-030-33368-3. DOI: https://doi.org/10.1007/978-3-030-33369-0_14

”industrial applications has shown how it is difficult to benefit from studies of this type, carried out on real raw data from operating machinery” - such difficulties might be also specified for other kind of research e.g. for transportation. Therefore literature review might be deepen, especially that it consists of machinery used in rail transport as well.

For future references prediction models might be used in Authors’ research - these are reviewed e.g. in Czwajda L., Kosacka-Olejnik M., Kudelska I., Kostrzewski M., Sethanan K., Pitakaso R., 2019, Application of prediction markets phenomenon as decision support instrument in vehicle recycling sector, LogForum, Vol. 15, Issue 2, pp. 265-278. DOI: 10.17270/J.LOG.2019.329

Figures with sensors allocations might be included.

The research methodology, as the literature review, was presented too briefly.

There are several parameters which are mentioned in the paper but their usage is probably hidden. Authors describe set of methods, techniques, parameters - all are given simply in a mess. Authors are asked to write a new version of the article according to the result and effect sequence.

„the study will continue in the next months, focusing on the treatment of the factors listed above” - Authors are asked to present new results as well.

The 4th section is not discussion - it is simply summary. The last section should be developed.